# Differential ripple propagation along the hippocampal longitudinal axis

Roberto De Filippo[1]*, Dietmar Schmitz[1,2,3,4,5]*

[1]Charité Universitätsmedizin Berlin, corporate member of Freie Universität Berlin, Humboldt-Universität zu Berlin, and Berlin Institute of Health; Neuroscience Research Center, Berlin, Germany; [2]German Center for Neurodegenerative Diseases (DZNE) Berlin, Berlin, Germany; [3]Charité-Universitätsmedizin Berlin, corporate member of Freie Universität Berlin, Humboldt-Universität Berlin, and Berlin Institute of Health, Einstein Center for Neuroscience, Berlin, Germany; [4]Charité-Universitätsmedizin Berlin, corporate member of Freie Universität Berlin, Humboldt-Universität Berlin, and Berlin Institute of Health, NeuroCure Cluster of Excellence, Berlin, Germany; [5]Humboldt-Universität zu Berlin, Bernstein Center for Computational Neuroscience, Berlin, Germany

**Abstract** Hippocampal ripples are highly synchronous neural events critical for memory consolidation and retrieval. A minority of strong ripples has been shown to be of particular importance in situations of increased memory demands. The propagation dynamics of strong ripples inside the hippocampal formation are, however, still opaque. We analyzed ripple propagation within the hippocampal formation in a large open-access dataset comprising 267 Neuropixel recordings in 49 awake, head-fixed mice. Surprisingly, strong ripples (top 10% in ripple strength) propagate differentially depending on their generation point along the hippocampal longitudinal axis. The septal hippocampal pole is able to generate longer ripples that engage more neurons and elicit spiking activity for an extended time even at considerable distances. Accordingly, a substantial portion of the variance in strong ripple duration ($R^2 = 0.463$) is explained by the ripple generation location on the longitudinal axis, in agreement with a possible distinctive role of the hippocampal septal pole in conditions of high-memory demand. Moreover, we observed that the location of the ripple generation has a significant impact on the spiking rate modulation of different hippocampal subfields, even before the onset of the ripple. This finding suggests that ripple generation location plays a crucial role in shaping the neural activity across the hippocampus.

*For correspondence:
roberto.de-filippo@charite.de
(RDF);
dschmitz-office@charite.de (DS)

**Competing interest:** The authors declare that no competing interests exist.

## Editor's evaluation

The findings of this manuscript were considered valuable, with theoretical or practical implications for a subfield. The strength of the evidence was convincing, in line with current state-of-the-art, with good support for the claims.

## Introduction

Hippocampal ripples are brief oscillatory events detected in the local field potential (LFP) of the hippocampal formation, these events correspond to the synchronized depolarization of a substantial number of neurons in various hippocampal subregions (*Hulse et al., 2016*; *Ylinen et al., 1995*). An higher ripple incidence during memory encoding is associated with superior recall performance (*Norman et al., 2019*), furthermore, ripple incidence is increased during successful memory retrieval (*Carr et al., 2011*; *Vaz et al., 2019*). Ripples are also involved in memory consolidation both in awake

and sleep conditions (*Girardeau et al., 2009*; *Jadhav et al., 2012*; *Roux et al., 2017*; *Sirota et al., 2003*), disrupting awake ripples during learning causes a persisting performance degradation, the same effect can be achieved by silencing ripples during post-learning sleep. Accordingly, ripples are considered to play a crucial role in memory processes and reorganization of memory engrams (*Buzsáki, 2015*; *Davidson et al., 2009*; *Diba and Buzsáki, 2007*; *Dragoi and Tonegawa, 2011*; *Foster and Wilson, 2006*; *Girardeau et al., 2009*; *Girardeau and Zugaro, 2011*; *Pfeiffer and Foster, 2015*; *Takahashi, 2015*; *Xu et al., 2019*). Ripples duration exhibits a skewed distribution with only a minority of long-duration ripples (>100 ms). The fraction of long-duration ripples, ripple amplitude, and within-ripple firing rate of both excitatory and inhibitory neurons are increased in both novel contexts and memory-demanding tasks (*Fernández-Ruiz et al., 2019*). Reducing ripple duration artificially causes a degraded working memory performance (*Fernández-Ruiz et al., 2019*; *Jadhav et al., 2012*) and, on the contrary, prolongation induced by optogenetic activation has a beneficial effect (*Fernández-Ruiz et al., 2019*). Importantly, the artificial recruitment of additional neurons seems to be constrained by pre-existing resting potential dynamics (*Noguchi et al., 2022*). Hippocampal–neocortical interactions, suggested to be important for memory consolidation (*Gais et al., 2007*; *Klinzing et al., 2019*; *Tukker et al., 2020*), are increased specifically during long-duration compared to short-duration ripples (*Ngo et al., 2020*).

Ripple amplitude and duration are significantly correlated (*Patel et al., 2013*; *Tong et al., 2021*), moreover, they are both related to the amount of underlying spiking activity (*Khodagholy et al., 2017*; *Tong et al., 2021*). It is possible to combine ripple strength and amplitude by considering the area of the high-pass filtered envelope (ripple strength).

These results point at a specific role of strong ripples (ripples with high strength) in situations of high mnemonic demand and are consistent with a possible power law distribution where a minority of ripples is responsible for a substantial part of memory requirements. For this reason, it is of interest to identify the possible electrophysiological peculiarities of this subgroup of ripples. Do strong ripples propagate differently compared to common ripples? Are strong ripples generated homogeneously along the hippocampal longitudinal axis? Do ripples have a preferred longitudinal directionality? In this study, we focused our attention on ripples generation and propagation within the hippocampal formation. Hippocampal connectivity with cortical and subcortical areas varies considerably along the longitudinal axis (*Fanselow and Dong, 2010*; *Moser and Moser, 1998*) and gene expression, as well, exhibits both gradual and discrete transitions along the same axis (*Strange et al., 2014*; *Vogel et al., 2020*). Consequently, the hippocampus is considered to be functionally segmented along its long axis. The different connectivity contributes to explain the functional organization gradient between a predominantly spatio-visual (septal pole) and emotional (temporal pole) processing. Ripples generated in the septal and temporal hippocampal pole have already been shown to be temporally independent and able to engage different neuron subpopulations, even in the same downstream brain area (*Sosa et al., 2020*). Consequentially, a heterogeneous ripple generation chance along the longitudinal axis most probably has an impact on the frequency with which different brain areas and neurons subgroups are activated by ripples. Our work is based on a dataset provided by the Allen Institute (*Siegle et al., 2021*), this dataset enabled us to study comprehensively ripples features across the septal half of the hippocampus. Previous studies have looked at ripple propagation along the longitudinal axes of the hippocampus (*Kumar and Deshmukh, 2020*; *Patel et al., 2013*), however, the size of this dataset made it possible to unveil propagation details previously overlooked.

## Results

### Distance explains most of the ripple strength correlation variability

We studied ripple propagation along the hippocampal longitudinal axis in an open-access dataset provided by the Allen Institute. We analyzed the LFP signals across the visual cortex, hippocampal formation, and brain stem (*Figure 1—figure supplement 1*) simultaneous to ripples detected in the CA1 of 49 animals (average session duration = 9877.4 ± 43.1 s, average ripple incidence during non-running epochs = 2.49 ± 0.12 per 10 s). Ripples (*n* ripples = 120,462) were detected on the CA1 channel with the strongest ripple activity. Ripple strength (∫Ripple) was calculated as the integral of the filtered LFP envelope between the start and end points for every detected ripple. Ripple strength and duration are highly correlated in each session (mean *r* = 0.87 ± 0.005, *Figure 1—figure supplement*

*2*). Notably ripple strength correlates significantly better with the hippocampal population spiking rate on a ripple-to-ripple basis compared to ripple duration alone (p-value = 4.31e−11, *Figure 1—figure supplement 3*). Clear ripples were observed uniquely in the hippocampal formation (CA1, CA2, CA3, DG, SUB, and ProS). Likewise, ripple-induced voltage deflections (RIVD, integral of the unfiltered LFP envelope) were also noticeably stronger in hippocampal areas (*Figure 1—figure supplement 4B–F*). Ripple strength was noticeably irregular in single sessions both across time and space, even within the CA1 region (*Figure 1—figure supplement 4C*). We focused on the variability in ripple strength across pairs of CA1 recording locations with clear ripple activity (*n* CA1 pairs = 303, *n* sessions = 46). Correlation of ripple strength across different CA1 regions was highly variable (*Figure 1A–C*) with a lower and upper quartiles of 0.66 and 0.87 (mean = 0.76, standard error of the mean = 0.01). Distance between recording location could explain the majority (57.6%) of this variability (*Figure 1B*) with the top and bottom quartiles of ripple strength correlation showing significantly different average distances (*Figure 1C, D*). Given the correlation variability we asked how reliably a ripple can travel along the hippocampal longitudinal axis. To answer this question, we looked at ripples lag in sessions that included both long-distance (>2126.66 μm) and short-distance (<857.29 μm) CA1 recording pairs (*n* sessions = 32, *n* CA1 pairs = 64, *Figure 1E*). Reference for the lag analysis was always the most medial recording location in each pair. Almost half of the ripples in long-distance pairs (49.3 ± 2.2%) were detected in both locations (inside a 120-ms window centered on ripple start at the reference location). Unsurprisingly short-distance pairs showed a more reliable propagation (69.59 ± 3.51%). Moreover, lag between long-distance pairs had a much broader distribution (*Figure 1F*) and a significantly bigger absolute lag (*Figure 1G*). Neither high nor short-distance pairs showed clear directionality (lag long distance = −1.14 ± 0.64 ms, lag short distance = −0.5 ± 0.41 ms). Looking at the relationship between lag and ripple strength in long-distance pairs, however, an asymmetric distribution was apparent (*Figure 1F* top), suggestive of a possible interaction between these two variables: stronger ripples appear to be predominantly associated with positive lags (i.e. ripples moving medial→lateral). To further investigate this relationship we divided ripples into two groups: strong (top 10% ripple strength per session at the reference location) and common (remaining ripples). The vast majority of the variance in 3D distance between recording locations was explained by the distance on the medio-lateral (M-L) axis alone (*R*² = 0.899). To simplify our analysis we therefore focused only on this spatial dimension. The septal half of the hippocampus was therefore divided along the M-L axis in three sections with equal number of recordings: medial, central, and lateral (*Figure 1—figure supplement 5*). Strong ripples identified in the medial section, in opposition to common ripples, showed a markedly positive lag (lag = 17.83 ± 1.02 ms) indicative of a preferred medial→lateral traveling direction (*Figure 1H* top). Surprisingly, the same was not true for strong ripples identified in the lateral section (lag = 3.62 ± 1.05 ms, *Figure 1I*). Strong and common ripples lags were significantly different between medial and lateral locations both in common and strong ripples. A biased direction of propagation can be explained by an unequal chance of ripple generation across space. We can assume that selecting strong ripples we are biasing our focus toward ripples whose generation point (seed) is situated nearby our reference location, this would contribute to explain the unbalanced lag. This notion would, however, fail to explain the different directionality we observed between strong ripples in medial and lateral locations. This hints at a more complex situation.

## Ripples propagates differentially along the hippocampal longitudinal axis

To analyze the propagation of ripples along the hippocampal longitudinal axis we focused on sessions from which ripples were clearly detected in at least two different hippocampal sections at the same time (*n* = 41). We followed the propagation of strong and common ripples detected in the reference location across the hippocampus (*Figure 2A, B*) and built an average spatio-temporal propagation map per session (*Figure 2C*). Strong and common ripples in the medial section showed a divergent propagation pattern: strong ripples traveling medio→laterally and common ripples traveling in the opposite direction (*Figure 2D, E*). Ripples detected in the lateral section did not show such strikingly divergent propagation (*Figure 2F, G*) whereas, in the central section, the propagation was divergent only laterally and not medially (*Figure 2H, I*). This peculiar propagation profile suggests a not previously described underlying directionality along the hippocampal longitudinal axis and can be possibly explained by a spatial bias in strong ripples generation. To understand the mechanism underlying such

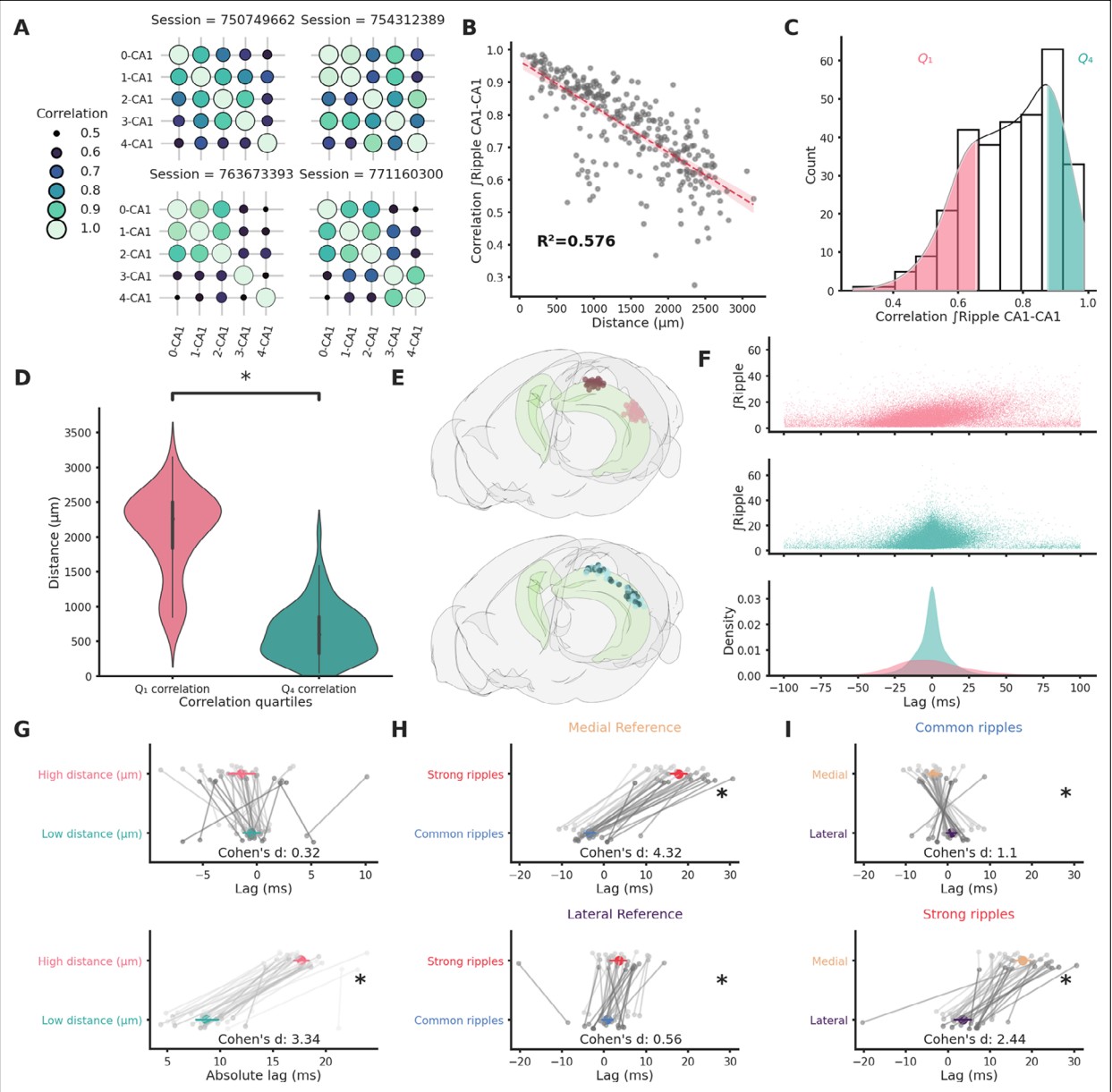

**Figure 1.** Ripple strength correlation depends significantly on distance. (**A**) Correlation matrices showing the variabilty of ripple strength correlation between pairs of recording sites located in different CA1 locations in four example sessions. The number on the x- and y-axis labels indicates the probe number. Probes are numbered according to the position on the hippocampal longitudinal axis (0 is the most medial probe). (**B**) Scatter plot and linear regression showing the relationship between distance and correlation strength. Distance between recording sites explains 0.576% of the variability in correlation of ripple strength. (**C**) Ripple strength correlation distribution. Pink represents bottom 25% ($<Q_1$) and blue top 25% ($>Q_4$). (**D**) Violin plots showing that the top and bottom correlation quartiles show significantly different distance distributions ($Q_1$: 2077.57 ± 68.68 μm, $Q_4$: 633.56 ± 44.02 μm, p-value = 4.00e−23, Mann–Whitney $U$ test). (**E**) Top: Rendering of the long-distance (top) and short-distance (bottom) CA1 pairs, dark circles are the reference locations in each pair. (**F**) Top and middle: scatter plots showing the relationship between ripple strength (at the reference location) and lag for long-distance (top, n ripples = 31,855) and short-distance (middle, n ripples = 52,858) pairs. Bottom: Kernel density estimate of the lags of long-distance (pink) and short-distance (turquoise) pairs. (**G**) Lag (top) and absolute lag (bottom) comparison between long- and short-distance pairs (top: long distance = −1.47 ± 0.63 ms, short distance = −0.51 ± 0.4 ms, p-value = 2.03e−01, Student's t-test; bottom: long distance = 17.69 ± 0.38 ms, short distance = 8.69 ± 0.56 ms, p-value = 6.58e−20, Student's t-test, asterisks mean p-value < 0.05). (**H**) Lag comparison in long-distance pairs between common and strong ripples with reference located inthe medial (top) or lateral hippocampal section (bottom) (top: strong ripples = 17.83 ± 1.02 ms, common ripples = −3.27 ± 0.68 ms, p-value = 2.28e−25, Student's t-test, bottom: strong ripples = 3.62 ± 1.05 ms, common ripples = 0.88 ± 0.66 ms, p-value = 3.00e−02, Student's t-test, asterisks mean p-value < 0.05). (**I**) Lag comparison in long-distance pairs between ripples with reference located in the medial and lateral sections in common (top) or strong ripples (bottom) (top: medial reference = −3.27 ± 0.68 ms, lateral reference = 0.88 ± 0.66 ms,

*Figure 1 continued*

p-value = 4.30e−05, Student's *t*-test, bottom: strong ripples = 17.83 ± 1.02 ms, common ripples = 3.62 ± 1.05 ms, p-value = 4.30e−05, Student's *t*-test, asterisks mean p-value < 0.05).

The online version of this article includes the following figure supplement(s) for figure 1:

**Figure supplement 1.** Spatial coordinates of all recorded brain regions.

**Figure supplement 2.** Correlation between ripple duration and strength per session.

**Figure supplement 3.** Comparison between correlation of ripple strength and duration with underlying spiking.

**Figure supplement 4.** Ripple-associated local field potential (LFP) responses are predominantly observed in hippocampal structures.

**Figure supplement 5.** Hippocampal sections.

difference in propagation we examined the location of the seed for each ripple in sessions in which ripples were clearly detected in every hippocampal section (n sessions = 25). While we found no differences in the number of ripples detected in each hippocampal section (p-value = 0.55, Kruskal–Wallis test), we observed differences regarding ripple generation. In common ripples, regardless of the reference location, most ripples started from the lateral section (*Figure 3A* left). On the other hand, strong ripples displayed a more heterogeneous picture (*Figure 3A* right). We identified two principles relative to strong ripples generation: In all hippocampal sections, the majority of strong ripples are locally generated, and a greater number of strong ripples is generated medially than laterally. Looking at the central section we can appreciate the difference between the number of strong ripples generated medially and laterally (*Figure 3A* right, mean medial = 36.83 ± 2.66%, mean lateral = 20.55 ± 2.04%, p-value = 3e−05, pairwise Tukey test). Strong and common ripples had significantly different seed location profiles only in the medial and central section, not in the lateral section (*Figure 3B*). These seed location profiles contribute to explain the propagation idiosyncrasies: major unbalances in seeds location cause propagation patterns with clear directionality, on the contrary, lag measurements hovering around zero are the result of averaging between two similarly numbered groups of ripples with opposite direction of propagation. Notably, propagation speed did not change depending on the seed location (*Figure 3—figure supplement 1*) and the antero-posterior (A-P) axis did not explain a considerable amount of variability in lag ($R^2$ medial = 0.0037, p-value = 4.48e−01, $R^2$ lateral = 0.0698, p-value = 1.74e−03), in contrast with the M-L axis ($R^2$ medial = 0.6648, p-value = 2.20e−38, $R^2$ lateral = 0.4989, p-value = 3.75e−22). The reason why strong ripples are only in a minority of cases generated in the lateral section remains nevertheless unclear. Using a 'strength conservation index' (SCI), we measured the ability of a ripple to retain its strength during propagation (a ripple with SCI = 1 is in the top 10% in all hippocampal sections). We observed that ripples generated laterally were effectively less able to retain their strength propagating toward the medial pole (*Figure 3—figure supplement 2*). This result is not simply explained by differences in ripple strength along the M-L axis, as no such gradient was observed ($R^2$=0.0012, *Figure 3—figure supplement 3*). Curiously, ripple amplitude showed a weak trend in the opposite direction (r = 0.27, p-value = 1.62e−04), with higher amplitude ripples in the lateral section (*Figure 3—figure supplement 4*).

## The hippocampal medial pole can generate longer ripples able to better engage neural networks

To understand the reason behind the differential propagation we focused uniquely on the central section, here it was possible to distinguish between ripples generated laterally or medially ('lateral ripples' and 'medial ripples'). We focused solely on ripples generated in the lateral and medial section, discarding the ones generated in the central section. We included in the analysis sessions in which ripples were clearly detected in each hippocampal section and with at least 100 ripples of each kind (n sessions = 24). We looked at spiking activity associated with these two classes of ripples in the hippocampal formation across the M-L axis (n clusters per session = 650.42 ± 33.16, *Figure 4A–C*). To compare sessions, we created interpolated maps of the difference between spiking induced by medial and lateral ripples (*Figure 4D*). Immediately following ripple start (0–50 ms, 'early phase') spiking was predictably influenced by ripple seed proximity: in the lateral section lateral ripples induced more spiking (indicated by the blue color), whereas in the medial section medial ripples dominated (indicated by the red color). Surprisingly, in the 50- to 120-ms window post-ripple start (late phase), medial ripples could elicit significantly higher spiking activity than lateral ripples along the entire M-L axis

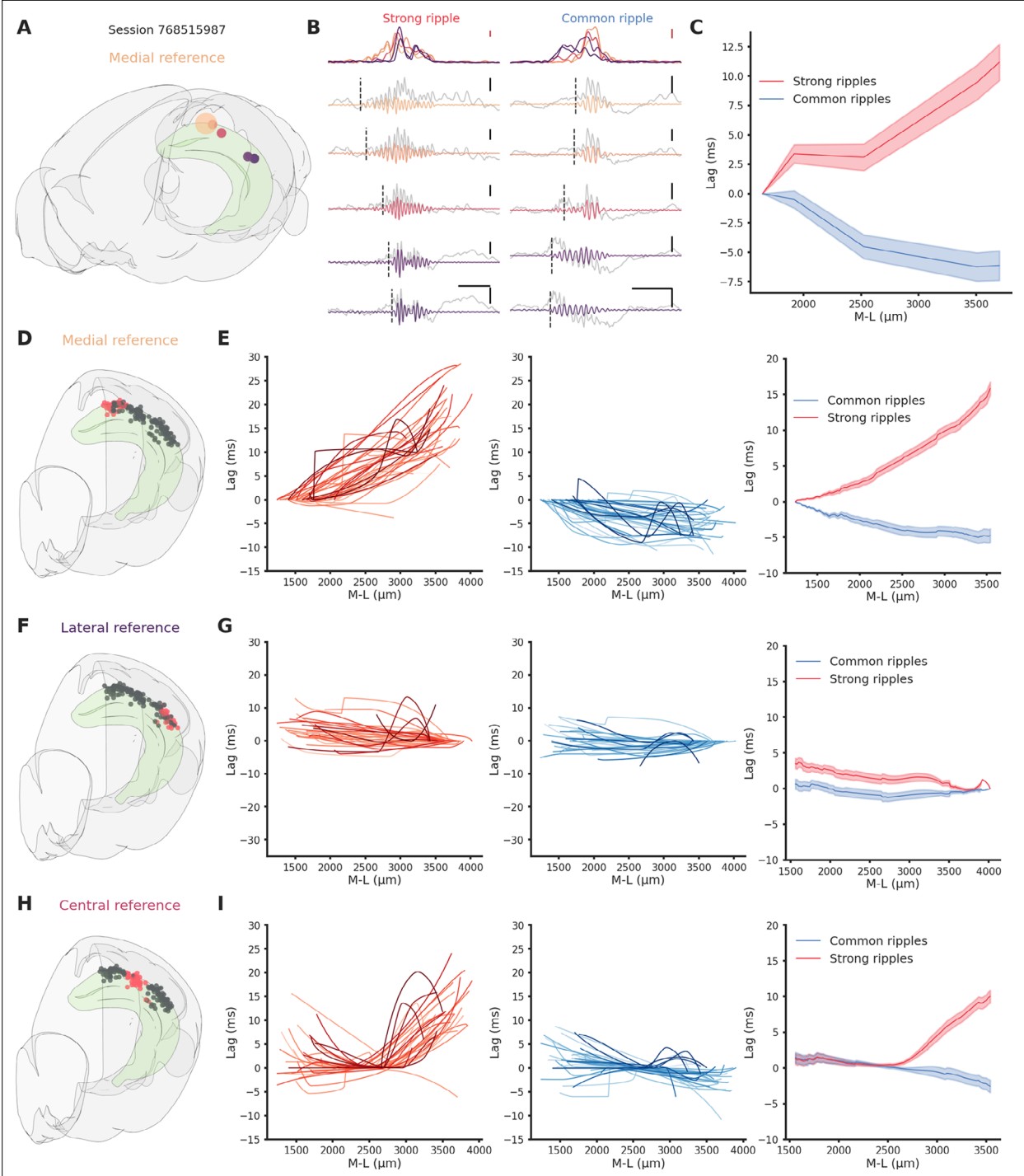

**Figure 2.** Direction-dependent differences in ripple propagation along the hippocampal longitudinal axis. (**A**) Recording locations for session 768515987. Circles colors represent medio-lateral location. Bigger circle represents the reference location. (**B**) Example propagation of a strong (left column) and common (right column) ripple across the different recording location from session 768515987, each filtered ripple is color coded according to A. Gray traces represent raw local field potential (LFP) signal. Dashed vertical line represents the start of the ripple. In the top row the ripple envelope across all locations. Black scale bars: 50 ms, 0.5 mV. Red scale bars: 0.1 mV. (**C**) Average propagation map of strong and common ripples in session 768515987 across the medio-lateral axis. (**D**) Recording locations relative to E. Red circles represent the reference locations across all sessions (*n* sessions = 41), black circles represent the remaining recording locations. (**E**) Left: Medio-lateral propagation of strong ripples, each line represents the average of one session. Middle: Medio-lateral propagation of common ripples, each line represents the average of one session. Right: Average propagation map across sessions of strong and common ripples. Reference locations are the most lateral per session. (**F**) Same as D. (**G**) Same as E. Reference locations are the most lateral per session. (**H**) Same as D. (**I**) Same as E. Reference locations are the most central per session.

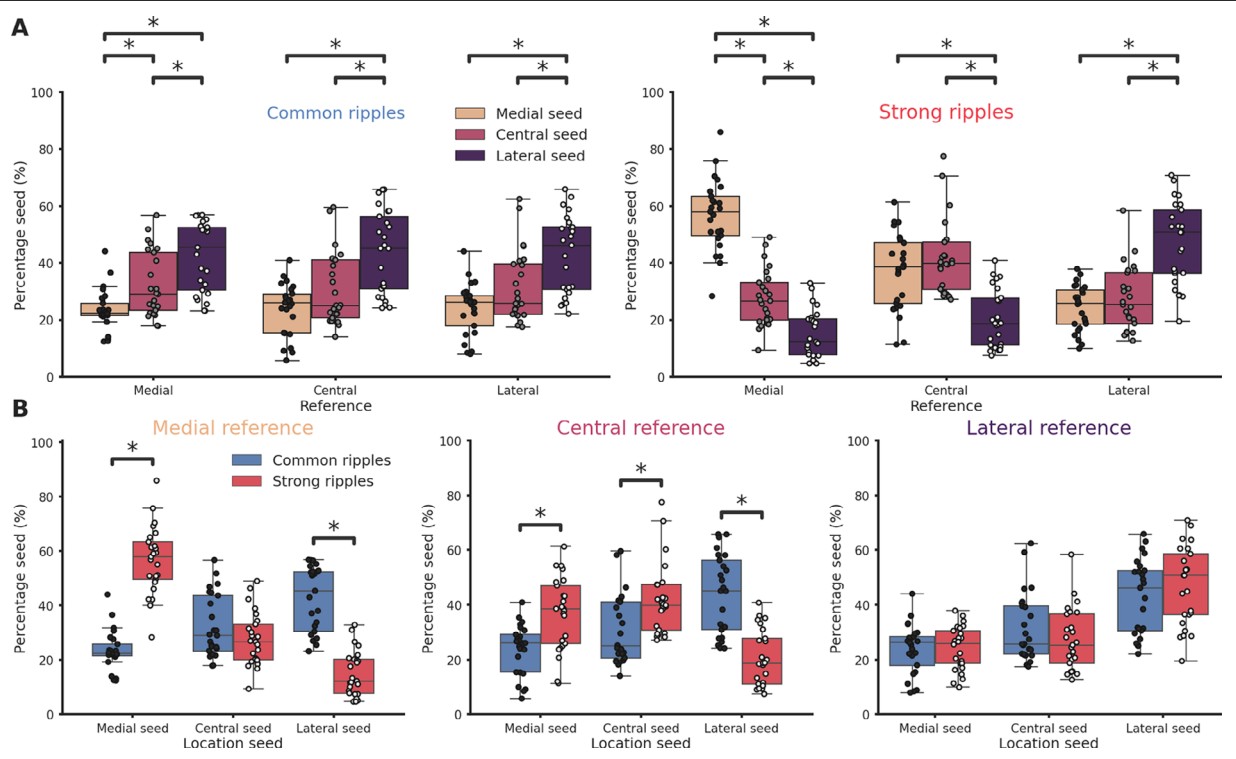

**Figure 3.** Ripples generation differences along the hippocampal longitudinal axis. (**A**) Ripple seed location comparison between the three reference locations in common ripples (left) and strong ripples (right). Majority of common ripples seeds are located in the lateral hippocampal section regardless of the reference location (medial reference/lateral seed = 42.43 ± 2.45%, central reference/lateral seed = 43.77 ± 2.9%, lateral reference/lateral seed = 42.83 ± 2.75%). Strong ripples are mainly local (medial reference/medial seed = 56.78 ± 2.48%, central reference/central seed = 41.74 ± 2.58%, lateral reference/lateral seed = 46.76 ± 2.89%). (**B**) Ripple seed location comparison between strong and common ripples using a medial (left), central (center), or lateral reference (right). Asterisks mean p < 0.05, Kruskal–Wallis test with pairwise Mann–Whitney post hoc test.

The online version of this article includes the following figure supplement(s) for figure 3:

**Figure supplement 1.** Spatio-temporal lag maps of locally and not locally generated ripples.

**Figure supplement 2.** Strength conservation in medially and laterally generated ripples.

**Figure supplement 3.** Spatial location does not influence ∫Ripple.

**Figure supplement 4.** Spatial location does not influence ripple amplitude.

(*Figure 4E*). Dividing clusters in putative excitatory and inhibitory using the waveform duration we observed the same effect in both types of neurons (*Figure 4—figure supplement 1*). In accordance with this result, we found that the medial hippocampal section is able to generate longer ripples (*Figure 4F*). An important portion of the variance in ripple duration is indeed explained by location on the M-L axis both in common ($R^2 = 0.133$) and especially in strong ripples ($R^2 = 0.463$). The observed extended spiking could be due to a increased number of neurons participating in the ripple, to a higher spiking rate per neuron or a combination of these two elements. Fraction of active neurons and spiking rate were both significantly higher in medial ripples (*Figure 4—figure supplement 2*). Focusing only on the late phase the difference in fraction of active neurons per ripples between medial and lateral ripples was even more striking (Cohen's $d$ = 1.7, *Figure 4G*). Inversely, in the early phase, lateral ripples could engage more neurons, although, the effect size was much smaller (Cohen's $d$=0.39). The same result was found in relation to the spiking rate, medial ripples caused a significant and considerable increase in spiking rate in the late phase (Cohen's $d$ = 1.75, *Figure 4H*). Dividing again the clusters into putative excitatory and inhibitory, significant differences between medial and lateral ripples were present only in the late phase. Spiking frequency and number of engaged neurons were significantly higher in medial ripples both in putative excitatory and inhibitory clusters (*Figure 4—figure supplement 3*). In summary, the prolonged spiking observed in medial ripples was caused both by an increased number of engaged neurons and a higher spiking rate per cell, both

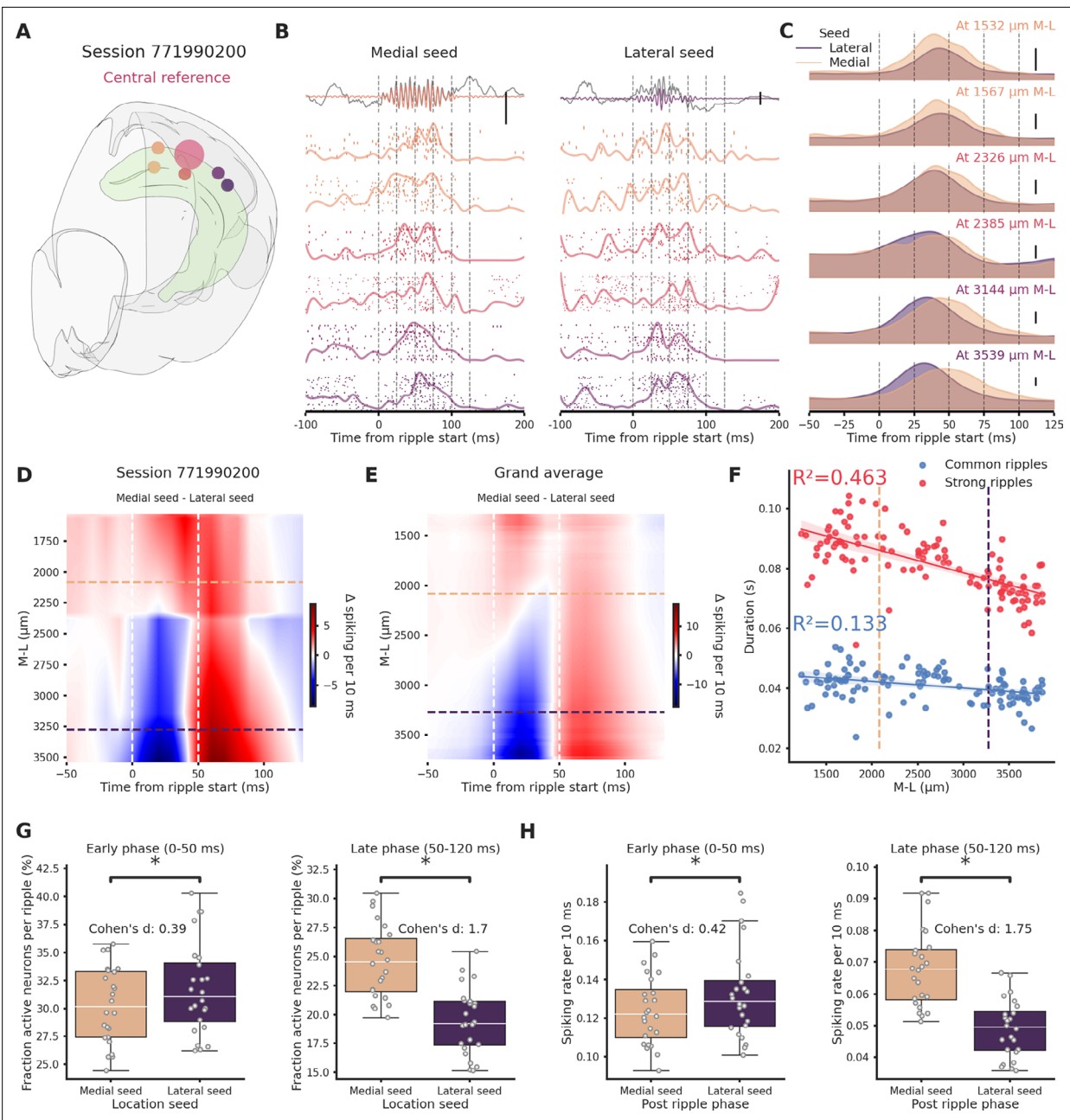

**Figure 4.** Ripples traveling in the medio→lateral direction show prolonged network engagement. (**A**) Recording location for session 771990200. Circles colors indicate medio-lateral (M-L) location. Bigger circle represents the reference location. (**B**) Spiking activity across the hippocampal M-L axis associated with a ripple generated medially (left column) or laterally (right column) across the different recording location from session 771990200. Spike raster plot and normalized density are plotted at each M-L location. In the top row filtered ripple, gray traces represent raw local field potential (LFP) signal. All plots are color coded according to A. Scale bar: 0.5 mV. (**C**) Kernel density estimates of the average spiking activity across different M-L locations and between seed type. Scale bar: 5 spikes per 10 ms. (**D**) Interpolated heatmap of the difference between medially and laterally generated ripple-induced spiking activity in session 771990200. Vertical dashed lines represent borders between early and late post-ripple start phases. Horizontal dashed lines represent the spatial limits of the hippocampal sections. (**E**) Grand average of the differences between medially and laterally initiated ripple-induced spiking activity across 24 sessions. Vertical dashed lines represent borders between early and late post-ripple start phases. Horizontal dashed lines represent the spatial limits of the hippocampal sections. (**F**) Regression plot between M-L location and ripple duration in common and strong ripples. Horizontal dashed lines represent the spatial limits of the hippocampal sections. (**G**) Average fraction of active neurons in medial (pink) and lateral (purple) ripples. Early/medial seed = 0.3 ± 0.69, early/lateral seed: 31.72 ± 0.84, p-value = 3.23e−05, Student's t-test; late/medial seed = 24.57 ± 0.64, late/lateral seed = 19.44 ± 0.58, p-value = 4.09e−07, Student's t-test. Asterisks mean p-value < 0.05. (**H**) Average spiking rate medial (pink) and lateral (purple) ripples. Early/medial seed = 0.12 ± 0.004, early/lateral seed = 0.13 ± 0.005, p-value = 1.35e−04, Student's t-test; late/medial seed = 0.07 ± 0.002, late/lateral seed = 0.05 ± 0.002, p-value = 1.24e−12, Student's t-test. Asterisks mean p-value < 0.05.

*Figure 4 continued on next page*

*Figure 4 continued*

The online version of this article includes the following figure supplement(s) for figure 4:

**Figure supplement 1.** Differential spiking of hippocampal neurons between different conditions.

**Figure supplement 2.** Spiking rate and fraction of active neurons are significantly higher in medial ripples.

**Figure supplement 3.** Spiking rate and fraction of active neurons are increased in the late phase post-ripple start in medial ripples both in putative excitatory and inhibitory neurons.

**Figure supplement 4.** Units features in medial and lateral sections.

in putative excitatory and inhibitory neurons. The disparity in network engagement can possibly be in part explained by electrophysiological differences across hippocampal sections (e.g. higher firing rate). We did not find differences in the number of firing neurons across the CA1 subfield (medial = 74.16, lateral = 69.66, p-value = 9.48e−01, Mann–Whitney $U$ test), we did, however, found differences in firing rate, waveform duration, and waveform shape (recovery slope and peak-trough ratio, *Figure 4—figure supplement 4*) in putative excitatory neurons. Firing rate and waveform duration in putative excitatory neurons exhibited, respectively, a left- and right-shifted distribution in the lateral section, reflecting lower firing rate and slower action potentials. Putative inhibitory interneurons in the lateral section showed a higher firing rate.

## Location of ripple seed is associated with different pattern of modulation across brain regions

To investigate how medial and lateral ripples affect various brain regions, we examined the modulation of spiking rate during ripples in individual clusters. A cluster was deemed modulated if it exhibited at least a 50% increase in spiking rate during either medial or lateral ripples. We found that clusters located in the thalamus (TH) and midbrain (MB) were hardly modulated (MB: 0.85 ± 0.71%, TH: 0.0 ± 0.0%), with baseline spiking rate explaining nearly all the variance in spiking rate during ripples (*Figure 5A*). Only a small fraction of cortical clusters were modulated (Isocortex: 5.26 ± 0.97%), in contrast, the majority of hippocampal (HPF) clusters showed ripple modulation (HPF: 88.57 ± 1.03%). The relationship between baseline spiking and spiking during ripples was similar in medial and lateral ripples, accounting for most of the observed variability (*Figure 5—figure supplement 1*). We found a modest difference in modulation of hippocampal clusters by medial and lateral ripples within a 120-ms window after the start of the ripple event (*Figure 5B*). A stronger effect was observed when isolating early and late phase. For example, in the 50- to 120-ms window medial ripples showed a notably stronger modulatory effect (*Figure 5—figure supplement 2A*). Significant differences were also observed in cortical clusters and in TH, with medial ripples inducing stronger modulation (*Figure 5—figure supplement 2B*). It is worth noting that the magnitude of this modulation is much smaller compared to the modulation observed in HPF. Clusters in the cortex that were modulated (by at least 25%, $n$ = 123/1240) by medial and lateral ripples were generally found in deeper layers (*Figure 5—video 1*), 32 clusters were modulated only by medial ripples and 25 only by lateral ripples. To understand the mechanism underlying the differences in modulation between medial and lateral ripples we focused on the early (0–50 ms) and late ripple phase (50–120 ms) in various hippocampal subfield (*Figure 5C*). In the early phase, we found a significantly stronger engagement of the dentate gyrus (DG), CA1 and CA3 areas and weaker engagement of the subiculum (SUB) by lateral ripples. The late phase analysis revealed further differences, as CA1, CA3, DG, prosubiculum (ProS), and SUB all displayed stronger modulation in response to medial ripples (*Figure 5D*). The clusters recorded across these various brain regions were not uniformly distributed along the M-L axis, which may have influenced the observed ripple modulation. For instance, it is conceivable that neurons located in closer proximity to the medial section may be more susceptible to the effects of medial ripples. To examine this hypothesis, we evaluated the degree to which variance in modulation could be attributed to the M-L axis position. We observed that during the early and late phases, the M-L position explained a substantial proportion of the variance in modulation in CA1 (early: 15.91%, late: 20.77%), but only a small proportion in CA3 (early: 0.48%, late: 0.34%), ProS (late: 4.03%), and SUB (early: 2.55%, late: 0.54%). In the DG subfield, the situation was mixed, with a notably greater impact of M-L position observed during the late phase (early: 0.76%, late: 10.93%). Additionally, we found that prior to the onset of ripples (20-ms window), medial ripple-induced stronger modulation in ProS and SUB, weaker

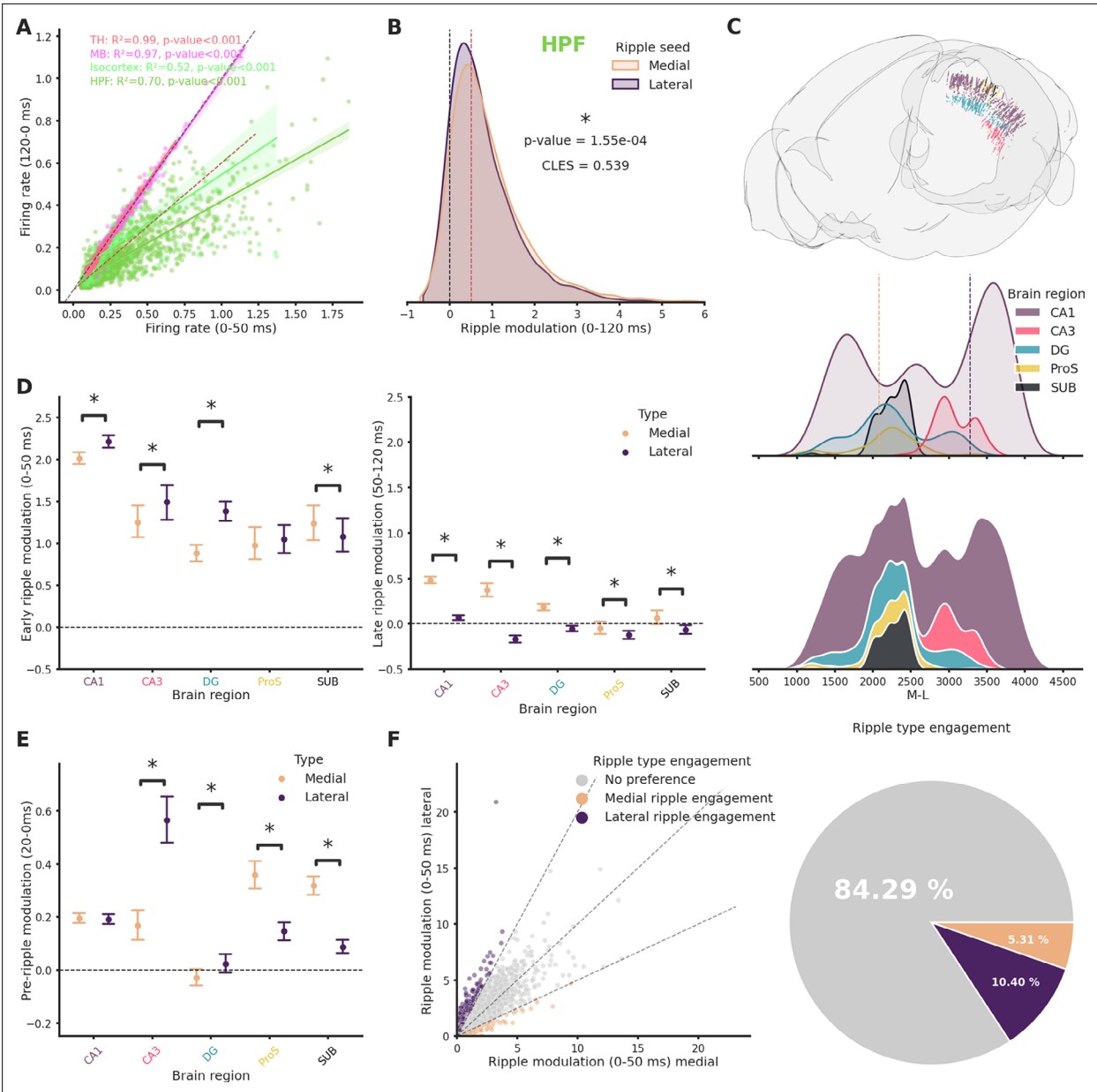

**Figure 5.** Ripple seed location influences the pattern of ripple modulation across various regions of the brain. (**A**) Relationship between baseline (120 ms before ripple start) and ripple (0–120 ms) firing rate for clusters recorded in Isocortex, hippocampal formation (HPF), thalamus (TH), and midbrain (MB). Spiking rates were calculated as the mean between responses to lateral and medial ripples. Dashed black line represents absence of any influence, dashed red line represents a 50% increased spiking rate. (**B**) Ripple modulation of hippocampal clusters in response to lateral and medial ripples. Dashed black line represents absence of any influence, dashed red line represents a 50% increased spiking rate. CLES = commn-language effect size. Wilcoxon signed-rank test. Asterisk mean p-value < 0.05. (**C**) Top: Rendering of all clusters recorded in the hippocampal formation color coded by subfield. Middle: Kernel density plot showing distribution of clusters along the medio-lateral (M-L) axis. Dashed lines represent medial and lateral limits. Bottom: Stacked kernel density plot showing distribution of clusters along the M-L axis. (**D**) Ripple modulation in response to lateral and medial ripples during the early (left) and late (right) ripple phase. Errorbar represents the standard error of the mean. Wilcoxon signed-rank test or Student's *t*-test (if normality established). Asterisks mean p-value < 0.05. (**E**) Ripple modulation in response to lateral and medial ripples before ripple start (20 ms). Errorbar represents the standard error of the mean. Wilcoxon signed-rank test or Student's *t*-test (if normality established). Asterisks mean p-value < 0.05. (**F**) Left: Relationship between modulation by lateral and medial ripples in hippocampal clusters. Dashed black line represents absence of difference and twofold differences in both directions. Right: Pie chart representing hippocampal clusters preference in ripple engagement.

The online version of this article includes the following video and figure supplement(s) for figure 5:

**Figure supplement 1.** Spiking rate modulation in medial and lateral ripples across brain regions.

**Figure supplement 2.** Ripple modulation density histograms.

*Figure 5 continued on next page*

**Figure supplement 4.** Pre-ripple modulation across hippocampal formation (HPF), Isocortex, midbrain (MB), and thalamus (TH).

**Figure supplement 5.** Clusters preference in ripple engagement by hippocampal subfields.

**Figure 5—video 1.** Cortical clusters showing ripple engagement.

modulation in DG and substantially weaker modulation in CA3 when compared to lateral ripples (*Figure 5E*). In this window, the M-L position explained only a minimal proportion of the variance in ProS (0.14%), SUB (2.65%), and CA3 (0.81%) but a more substantial proportion in DG (8.68%). In both the early and late ripple phases, medial ripples elicited significantly more modulation across multiple cortical regions (*Figure 5—figure supplement 3*). This effect was also observed before the onset of ripples (*Figure 5—figure supplement 4*). Furthermore, we investigated whether neurons exhibit a preference for medial or lateral ripples. To address this question, we identified clusters that exhibited at least a twofold modulation difference for either medial or lateral ripples (with absolute modulation of at least 50%). Our analysis revealed that the majority of clusters did not display a strong preference (84.29%), whereas 10.4% of clusters responded preferentially to lateral ripples and 5.31% to medial ripples (*Figure 5F*). Interestingly, our analysis also revealed that these preferences for lateral or medial ripples varied considerably across hippocampal subfields, with striking differences observed between the different subfields. Specifically, DG exhibited the highest proportion (28.15%) of neurons responsive to lateral ripples. In contrast, SUB displayed the highest proportion (11.68%) of neurons responsive to medial ripples (*Figure 5—figure supplement 5*). These results suggest a fundamental difference between medial and lateral ripples in terms of their engagement of brain regions, highlighting that these differences exist even before the ripples are detected. Notably, most of these differences are not explained by the location of the neuron along the M-L axis. This suggests that there are more fundamental differences at play beyond just spatial distance.

## Discussion

Our results show for the first time that strong ripples propagate differentially along the hippocampal longitudinal axis. This propagation idiosyncrasy can be explained by a specific ability of the hippocampal septal pole (medial section in our analysis) to produce longer ripples that better entrain the hippocampal network and spread across the longitudinal axis. It was previously observed that ripples located at the septal and temporal pole are generated independently from each other, in addition, despite the presence of connections within the hippocampal longitudinal axis (*van Strien et al., 2009*; *Witter, 2007*), in the vast majority of cases ripples do not propagate to the opposite pole (*Sosa et al., 2020*). In accordance with these results, we observed a strong effect of spatial distance on ripple strength correlation confirming a previous study (*Nitzan et al., 2022*): the strength correlation, predictably, was higher in CA1 pairs closer to each other. The effect of distance was also apparent on the ripple chance of propagation, only half of the ripples generated in the septal pole were detected additionally in the intermediate hippocampus (lateral section in our analysis). This chance is much higher compared to the ~3.7% reported regarding propagation between opposite poles (*Sosa et al., 2020*), it would be interesting to understand whether the temporal pole is also able to entrain the intermediate hippocampus in similar fashion or it is a peculiarity of the septal pole. A limitation of our work derives from the dataset being limited to the septal and intermediate hippocampus.

Ripples can arise at any location along the hippocampal longitudinal axis (*Patel et al., 2013*). Our analysis shows that ripples are, however, not homogeneously generated across space. We observed important differences between strong ripples and common ripples generation. Common ripples followed a gradient with higher generation probability in the intermediate section and lowest in the septal pole. Strong ripples, on the other hand, were mostly generated locally (i.e. a strong ripple detected in the medial section is most likely generated in the medial section itself). Furthermore, only rarely a strong ripple generated in the intermediate hippocampus is able to propagate toward the septal pole retaining its strong status (top 10%). Conversely strong ripples generated in the septal pole have a significantly higher chance of propagate longitudinally and still be in the top 10% in

terms of ripple strength. Notably, this is not consequence of a simple longitudinal gradient in ripple strength, indeed, we did not observe any difference in ripple strength along the longitudinal axis. Additionally, we show that ripples generated in the septal pole and in the intermediate hippocampus have a significantly different ability to engage hippocampal networks in the 50- to 120-ms window post-ripple start. Ripples generated in the septal pole activate more neurons, both excitatory and inhibitory, and, moreover, elicit an higher spiking rate per neuron. This prolonged network activation is reflected by the fact that the position on the longitudinal axis explains 13.3% and 46.3% of the variability in ripple duration in common and strong ripples, respectively. Consistent with a duration gradient along the longitudinal axis, the temporal hippocampus has been shown to produce shorter ripples both in awake and sleep conditions (*Sosa et al., 2020*).

What is the reason that enables the septal pole to generate longer ripples? There might be for example underlying electrophysiological differences between the septal and intermediate hippocampus. Upon closer examination of the electrophysiological features of the neurons, we were able to discern significant differences in the shape and duration of their waveform. We can hypothesize that slower action potentials and, consequentially, longer refractory periods hinder the ability to sustain protracted high frequency spiking. Accordingly, we found an increased firing rate and a smaller waveform duration in putative excitatory neurons of the septal pole. Moreover, putative inhibitory neurons in the septal pole showed reduced firing. These differences might contribute to explain the prolonged ripples observed in the septal pole. We can also speculate that the neuromodulatory inputs gradient, monoamine fibers have been shown to be stronger in the ventral part (*Strange et al., 2014*), might influence neurons responses. Serotonin (*ul Haq et al., 2016*; *Wang et al., 2015*), noradrenaline (*Novitskaya et al., 2016*; *Ul Haq et al., 2012*), and acetylcholine *Zhang et al., 2021* have all been shown to suppress ripples. In accordance with this, some ripples are coupled with a reduced activation of the locus coeruleus and the dorsal raphe nucleus in vivo (*Ramirez-Villegas et al., 2015*).

Ripples can be subdivided in different types according to the relationship between the hippocampal LFP and the ripple itself (*Ramirez-Villegas et al., 2015*). Intriguingly, these subtypes are associated with two different brain-wide networks, the first communicating preferentially with the associative neocortex and a second one biased toward subcortical structures. Moreover, these different types of ripples have been proposed to possibly fulfill different functional roles. Given the different input/output connectivity between septal, intermediate, and temporal hippocampus (*Fanselow and Dong, 2010*) we hypothesize that ripple generated at different points of the hippocampal longitudinal axis might as well have functional differences, with the longer ripples generated septally possibly able to combine the different kind of information processed in the distinct hippocampal sections and additionally relaying the integrated information back to the neocortex in accordance with the two-stage memory hypothesis (*Buzsáki, 1989*; *Diekelmann and Born, 2010*; *Marr, 1971*; *McClelland et al., 1995*; *Rasch and Born, 2007*).

Long-duration ripples have been shown to be of particular importance in situations of high-memory demand (*Fernández-Ruiz et al., 2019*), at the same time, previous studies highlighted the role of septal hippocampus in memory tasks and information processing (*Bradfield et al., 2020*; *Fanselow and Dong, 2010*; *Hock and Bunsey, 1998*; *Kheirbek et al., 2013*; *Maras et al., 2014*; *McGlinchey and Aston-Jones, 2018*; *Moser et al., 1993*; *Moser et al., 1995*; *Qin et al., 2021*; *Steffenach et al., 2005*). Our results can contribute to explain the specific role of septal hippocampus in memory-demanding tasks with its ability of generating particularly long ripples that are able to strongly engage networks in the entire top half of the hippocampal formation for an extended time.

Functional differences between ripples are supported by our finding that the ripple origination point has an influence on the engagement of various hippocampal subfields. Interestingly, these functional differences can be observed even before the onset of ripples. Our analysis revealed that DG, CA1, and CA3 subfields of the hippocampus were more strongly engaged during the early phase in response to medial ripples, the opposite was true for SUB. In the late phase, all subfields, including ProS, showed stronger modulation in response to medial ripples. Moreover, the analysis of pre-ripple activity showed that ProS and SUB exhibited stronger modulation in response to medial ripples, while DG and, especially, CA3 displayed weaker modulation compared to lateral ripples. These results indicate that CA3 may play a more critical role in initiating lateral ripples, in accordance with the traditional view that CA3 is the primary generator of ripples (*Buzsáki, 1986*; *Buzsáki, 1989*; *Csicsvari et al., 2000*). On the other hand, medial ripples demonstrate stronger engagement of ProS and SUB

subfields before ripple start, supporting the more recent hypothesis that output structures in the hippocampus are also capable of generating ripples (*Imbrosci et al., 2021*). The subicular complex has been suggested to play a role in the transfer of information from the hippocampus to other brain regions, such as the cortex (*Aggleton and Christiansen, 2015*; *Naber and Witter, 1998*). It is possible that the weaker engagement of the subiculum during lateral ripples may reflect a more localized processing of information within the hippocampus, while the stronger engagement of the subiculum during medial ripples may reflect a more global transfer of information to other brain regions. Differences in ripple initiation and engagement patterns can provide valuable insights into the mechanisms that underlie the dynamics of the hippocampal network. Gaining a better understanding of these differences may shed light on the functional significance of ripples in the brain and their role in memory consolidation and retrieval.

## Materials and methods

### Dataset

Our analysis was based on the Visual Coding – Neuropixels dataset of head-fixed recordings in awake mice provided by the Allen Institute and available at https://allensdk.readthedocs.io/en/latest/visual_coding_neuropixels.html. We excluded six sessions because of absence of recording electrodes in CA1 (session ids = 732592105, 737581020, 739448407, 742951821, 760693773, 762120172). Furthermore, one session was excluded (session id = 743475441) because of an artifact in the LFP time series (time was not monotonically increasing) and two other sessions (session ids = 746083955, 756029989) because of duplicated LFP traces (see https://github.com/RobertoDF/Allen_visual_dataset_artifacts). Our analysis was therefore focused on 49 sessions, average animal age = 119.22 ± 1.81. Sex: males $n = 38$, females $n = 11$. Genotypes: wt/wt $n = 26$, Sst-IRES-Cre/wt;Ai32(RCL-ChR2(H134R)_EYFP)/wt $n = 10$, Vip-IRES-Cre/wt;Ai32(RCL-ChR2(H134R)_EYFP)/wt $n = 7$, and Pvalb-IRES-Cre/wt;Ai32(RCL-ChR2(H134R)_EYFP)/wt $n = 6$. Average probe count per session = 5.73 ± 0.08. Average number of recording channels per session = 2129.45 ± 29.46. Probes in each session were numbered according to the position on the M-L axis, with probe number 0 being the most medial. Channels with ambiguous area annotations were discarded (e.g. HPF instead of CA1). We found a number of of small artifacts in a variety of sessions, all this timepoints were excluded from the analysis (for more information: https://github.com/RobertoDF/Allen_visual_dataset_artifacts). Each recording session had a length of ~3 hr. In each experiment, a series of visual stimuli were presented in front of the animal (gabors, drifting gratings, static gratings, natural images, movies, flashes). Mice did not undergo any training associated with these stimuli. Further details about data acquisition can be found at https://brainmap-portal-live-4cc80a57cd6e400d854-f7fdcae.divio-media.net/filer_public/80/75/8075a100-ca64-429a-b39a-569121b612b2/neuropixels_visual_coding_-_white_paper_v10.pdf. Visualization of recording locations was performed with brainrender (*Claudi et al., 2021*).

### Ripples detection

The LFP traces sampled at 1250 Hz were filtered using a sixth-order Butterworth bandpass filter between 120.0 and 250.0. Ripples were detected on CA1 LFP traces, the best channel (higher ripple strength) was selected by looking at the SD of the envelope of the filtered trace, if multiple SD peaks were present across space (possibly caused by sharp waves in stratum radiatum and ripple activity in stratum pyramidale) we subsequently looked at the channel with higher skewness, in this way we could reliably identify the best ripple channel. The envelope of the filtered trace was calculated using the Hilbert transform (scipy.signal.hilbert). Ripple threshold was set at 5 SDs. Start and stop times were calculated using a 2 SDs threshold on the smoothed envelope with window = 5 (pandas.Data-Frame.rolling) to account for ripple phase distortions. Ripple amplitude was calculated as the 90th percentile of the envelope.Ripple duration was limited at >0.015 and <0.25 s. Candidate ripples with starting times closer than 0.05 s were joined in a single ripple with peak amplitude being the highest between the candidates. We estimated power density of each candidate using a periodogram with constant detrending (scipy.signal.periodogram) on the raw LFP trace, we checked the presence of a peak >100 Hz, candidates not fulfilling this condition were discarded, this condition was meant to reduce the number of detected false positives. Ripple candidates detected during running epochs were discarded, an animal was considered to be running if his standardized speed was higher than the

10th percentile plus 0.06. Candidates were also discarded if no behavioral data was available. Code for the detection of ripples resides in 'Calculate_ripples.py'.

## Correlation and lag analysis

In each session, we uniquely used ripples from the CA1 channel with the strongest ripple activity, we looked at the LFP activity in all brain areas recorded in a window of 100.0-ms pre-ripple start and 200.0-ms post-ripple start, this broad windows account for possible traveling delays due to distance. For each brain area, we picked the channel with higher SD of the envelope of the filtered trace. For each ripple considered we calculated integral of the envelope of the filtered trace (∫Ripple) and the integral of the raw LFP (RIVD). After discarding channels with weak ripple activity (envelope variance <5), we computed the pairwise pearson correlation of the envelope traces of CA1 channels ( pandas.DataFrame.corr). For the lag analysis, we first identified pairs of CA1 that satisfied a distance requirements. Distance threshold were set at 25% (857.29 µm) and 75% (2126.66 µm) of the totality of distances. For each ripple detected in the reference channel we identifired the nearest neighbor in the other channel. The analysis was repeated after dividing ripples in strong (top 10% ∫Ripple) and common ripples (all remaining ripples) per session. Code for the correlation and lag analysis resides in 'Calculations_Figure_1.py'.

## Ripple spatio-temporal propagation maps and ripple seed analysis

The hippocampus was divided in three section with equal number of recordings. Channels with weak ripple activity (envelope variance <5) were discarded. Sessions with recording locations only in one hippocampal sections or with less than 1000 ripples in the channel with strongest ripple activity were discarded as well. For each ripple detected on the reference CA1 channel we identified ripples in other CA1 channels happening in a window of ±60.0 ms, this events were grouped together in a 'cluster'. If more than one event was detected on the same probe we kept only the first event. 'Clusters' were subsequently divided according to ∫Ripple on the reference electrode in strong and common ripples. Lag maps were result of averaging lags for each probe. Code for the calculations of propagation maps resides in 'Calculate_trajectories.py'.

## Ripple-associated spiking activity

We focused on sessions with clear ripple activity (envelope variance >5) in all three hippocampal sections and at least 100 ripples generated both medially and laterally. The reference was always placed in the central section, here it was possible. to identify ripples generated medially and laterally. We only considered ripples that were detected in at least half of the recording electrodes (in the code: "spatial engagment" >0.5). For each ripple we computed a histogram of spiking activity of regions belonging to the hippocampal formation (HPF) in a window of 0.5 s centered on the ripple start in each probe. We averaged all the computed histograms to create a spatial profile of spiking activity. To compare spiking activity between sessions we interpolated (xarray.DataArray.interp) the difference between medial and lateral ripple-induced spiking over space (this was necessary because probes in each session have different M-L coordinates) and time. We calculated the number of active cells (at least one spike) and spiking rate of each cluster per ripple in a window of 0.12 s starting from ripple start. We repeated the analysis separating the 0- to 50-ms and 50- to 120-ms post-ripple start windows. The degree of association between ripple lags and M-L or A-P axis was calculated using partial correlation (https://pingouin-stats.org/build/html/generated/pingouin.partial_corr.html), in this way we could remove the effect of the other axis.

## Units selection, electrophysiological features calculations, and ripple modulation

Clusters were filtered according to the following parameters: waveform peak-through ratio <5, ISI violations <0.5, amplitude cutoff <0.1, and presence ratio >0.1. For an explanation of the parameters see https://github.com/AllenInstitute/ecephys_spike_sorting/blob/master/ecephys_spike_sorting/modules/quality_metrics/README.md and https://brainmapportal-live-4cc80a57cd6e400d854-f7fdcae.divio-media.net/filer_public/80/75/8075a100-ca64-429a-b39a-569121b612b2/neuropixels_visual_coding_-_white_paper_v10.pdf. Firing rate was calculated on all clusters with presence ratio >0.1. Ripple modulation was calculated only for sessions with at least one recording in both the lateral

and medial section ($n$ = 24) and only in clusters with firing rate >0.1 spikes/s. Ripple modulation was calculated as (ripple spiking rate − baseline spiking rate)/baseline spiking rate.

## Acknowledgements

This study was supported by the German Research Foundation Deutsche Forschungsgemeinschaft (DFG), project 184695641 – SFB 958, project 327654276 – SFB 1315, Germany's Excellence Strategy – Exc-2049-390688087 and by the European Research Council (ERC) under the European Union's Horizon 2020 research and innovation programme (grant agreement no. 810580). We thank JT Tukker, N Maier for feedback on an early version of the manuscript and the members of the Schmitz lab for scientific discussion. We thank Willy Schiegel and Tiziano Zito for technical help with cluster computing. We thank Federico Claudi for support with brainrender. The authors declare that they have no competing interests.

## Additional information

### Funding

| Funder | Grant reference number | Author |
| --- | --- | --- |
| Deutsche Forschungsgemeinschaft | 184695641 - SFB 958 | Dietmar Schmitz |
| Deutsche Forschungsgemeinschaft | 327654276 - SFB 1315 | Dietmar Schmitz |
| Horizon 2020 - Research and Innovation Framework Programme | 810580 | Roberto De Filippo |
| NeuroCure Exzellenzcluster | Exc-2049-390688087 | Dietmar Schmitz |

The funders had no role in study design, data collection, and interpretation, or the decision to submit the work for publication.

### Author contributions

Roberto De Filippo, Conceptualization, Data curation, Software, Formal analysis, Supervision, Investigation, Visualization, Methodology, Writing – original draft, Project administration, Writing – review and editing; Dietmar Schmitz, Resources, Funding acquisition, Writing – review and editing

### Author ORCIDs

Roberto De Filippo ⓘ http://orcid.org/0000-0002-4085-9114
Dietmar Schmitz ⓘ http://orcid.org/0000-0003-2741-5241

### Decision letter and Author response

Decision letter https://doi.org/10.7554/eLife.85488.sa1
Author response https://doi.org/10.7554/eLife.85488.sa2

## Additional files

### Supplementary files

• MDAR checklist

### Data availability

All the code used to process the dataset is available at https://github.com/RobertoDF/De-Filippo-et-al-2022 (copy archived at swh:1:rev:2d4bf657fd87cd9483932cb47e65d5c7384a7d65). Pre-computed data structures can be downloaded at https://figshare.com/s/d1c93882d6438646dd64. All figures and text can be reproduced using code present in this repository, each number present in the text is directly linked to a python data structure. The original dataset is provided by the Allen Institute and available at https://allensdk.readthedocs.io/en/latest/visual_coding_neuropixels.html.

The following dataset was generated:

| Author(s) | Year | Dataset title | Dataset URL | Database and Identifier |
|-----------|------|---------------|-------------|--------------------------|
| De Filippo R | 2023 | All the code used to process the dataset | https://doi.org/10.6084/m9.figshare.20209913 | figshare, 10.6084/m9.figshare.20209913 |

The following previously published dataset was used:

| Author(s) | Year | Dataset title | Dataset URL | Database and Identifier |
|-----------|------|---------------|-------------|--------------------------|
| Siegle JH, Jia X, Durand S | 2021 | Visual Coding - Neuropixels | https://knowledge.brain-map.org/data/4YYLRZZGK82FQ85NIH8/summary | Allen Brain Map, 4YYLRZZGK82FQ85NIH8 |

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
