## [Editor Report]

The findings of this manuscript were considered valuable, with theoretical or practical implications for a subfield. The strength of the evidence was convincing, in line with current state-of-the-art, with good support for the claims.

---

## [Decision Letter]

**Decision letter after peer review:**

Thank you for submitting your article "Differential ripple propagation along the hippocampal longitudinal axis" for consideration by *eLife*. Your article has been reviewed by 3 peer reviewers, including Liset M de la Prida as Reviewing Editor and Reviewer #1, and the evaluation has been overseen by Laura Colgin as the Senior Editor. The following individuals involved in the review of your submission have agreed to reveal their identity: Manuel Valero (Reviewer #2); Kazumasa Tanaka (Reviewer #3).

Essential revisions:

We all agree there is value in the paper, but we need to clarify what is new as compared to previous papers in the field targeting similar questions. We also need to see further analysis and clarifications of results to improve the strength of evidence. In particular:

1) Further analysis and clarification should be provided on the proximodistal and longitudinal axes and how they relate to results on mediolateral effects. Please, consider rephrasing the title accordingly.

2) Separation of cell types to reinforce results on potential mechanisms.

3) To discuss how these data compare to previous reports.

Please, go through all the reviewers' comments to address these three main points carefully.

Hippocampal sectors should be more thoroughly addressed in support of mechanisms.

*Reviewer #1 (Recommendations for the authors):*

Specific comments:

– The definition of the longitudinal and transversal axis is a bit unclear. While the title refers to the longitudinal axis the overall focus in the manuscript is the M-L coordinates. It is unclear what is the relationship between M-L and the longitudinal (dorsoventral) and transverse axes (proximodistal). Are ripples generated laterally or medially reflecting longitudinal or traverse trends of hippocampal anatomy and physiology?

– The study remains a bit descriptive, except for the analysis of neuronal dynamics which aims at providing some mechanism. Yet, the level of insights remains poor and the analysis is confined to basic properties of unit waverfoms and rates. The paper will benefit from addressing some additional potential factors. For example, what microcircuit factors can account for the M-L organization of ripple propagation? Is CA3-CA1 coordination (and possible SUB-CA1 coordination) differently involved? Are ripples generated laterally or medially involving different coordination between different CA1 cell types and neurons in these regions? In addition, the interaction between pyr and interneurons, and/or the contribution of interneurons from different layers, may entail different subcircuits for ripples generated laterally or medially. This should be more thoroughly addressed in a separate figure to reinforce a touch on the mechanisms.

– One interesting finding is the different waveform duration and shapes of neurons along the M-L coordinates but it is unclear whether it refers to putative pyramidal cells or interneurons, whether all cells belong to the CA1 region or they rather refer to different hippocampal sectors (sup.Figure 13; no anatomical information, no raw examples provided). It is unclear how this observation relates to ripples of different origins.

*Reviewer #3 (Recommendations for the authors):*

– For better clarification of the nature of the collected data, the behavioral states of the animals (passive presentation of visual stimuli, head-fixed, etc) should be mentioned in the Dataset section in the Materials and methods.

– As described in the public review, the authors can provide more thoughts on the physiological impacts and or a particular role of the directional propagation of the ripples. However, this suggestion is optional.

– Kumar and Deshmukh report stronger ripple propagation along the proximodistal axis. I am wondering how much of the present finding is explained by the difference between the proximal vs distal CA1 because I see the Neurapixels data from the intermediate CA1 is dominated by the proximal part (Figure 1E). Additional analysis of differentially characterizing the ripple propagation along the proximodistal axis would strengthen the conclusion.

---

## [Author Response]

Reviewer #1 (Recommendations for the authors):Specific comments:– The definition of the longitudinal and transversal axis is a bit unclear. While the title refers to the longitudinal axis the overall focus in the manuscript is the M-L coordinates. It is unclear what is the relationship between M-L and the longitudinal (dorsoventral) and transverse axes (proximodistal). Are ripples generated laterally or medially reflecting longitudinal or traverse trends of hippocampal anatomy and physiology?

We apologize for the lack of explanation about the use of “longitudinal” and “M-L (medio-lateral)”. Our use of the adjective “longitudinal” is in line with the usage of (Henriksen et al., 2010); Strange et al. (2014), this is meant to refer to the long axis of the hippocampus, therefore combining all spatial axis (D-V, M-L and A-P). This is reflected by the curved arrow in Figure 2A in Strange et al. 2014.

Given the hippocampal formation shape (and considering only the top half of the hippocampus, as done in our study because of constraints in the dataset), the D-V (dorsal-ventral) and M-L axis are highly correlated (r = 0.75). This means that a lateral recording location will always be more ventral than a medial recording location. The M-L axis was chosen because it could explain the largest share of variance in the absolute distance between recording locations in CA1 (R^2^=0.90). D-V axis explained 78% of the variance and the A-P 28%. We added a panel (A) in supplementary figure 5 and a brief explanation in the Results section on why we choose the M-L axis:

“The vast majority of the variance in 3D distance between recording locations was…”

– The study remains a bit descriptive, except for the analysis of neuronal dynamics which aims at providing some mechanism. Yet, the level of insights remains poor and the analysis is confined to basic properties of unit waverfoms and rates. The paper will benefit from addressing some additional potential factors. For example, what microcircuit factors can account for the M-L organization of ripple propagation? Is CA3-CA1 coordination (and possible SUB-CA1 coordination) differently involved? Are ripples generated laterally or medially involving different coordination between different CA1 cell types and neurons in these regions? In addition, the interaction between pyr and interneurons, and/or the contribution of interneurons from different layers, may entail different subcircuits for ripples generated laterally or medially. This should be more thoroughly addressed in a separate figure to reinforce a touch on the mechanisms.

In response to this comment and comment #3 of reviewer #2 we have incorporated a detailed analysis of ripple modulation on a neuron level in each hippocampal subfield (Figure 5 and associated supplementary figures).

We observed significant differences in the modulation by medial and lateral ripples on hippocampal neurons, in particular, different hippocampal subfields displayed varying levels of modulation by lateral and medial ripples. We updated our discussion (last paragraph) with a new ending referring to these new results.

– One interesting finding is the different waveform duration and shapes of neurons along the M-L coordinates but it is unclear whether it refers to putative pyramidal cells or interneurons, whether all cells belong to the CA1 region or they rather refer to different hippocampal sectors (sup.Figure 13; no anatomical information, no raw examples provided). It is unclear how this observation relates to ripples of different origins.

We updated the analysis to include only CA1 units, we clarified this in the text. The average waveform for putative excitatory and inhibitory neurons in lateral and medial hippocampal sections has been added in the figure, following the reviewer #1 advice. In accordance with the advice from reviewer #2 the analysis of units features has been divided in putative excitatory and inhibitory neurons. In the discussion we try to interpret these results in the context of ripple propagation (“We can hypothesize that slower action potentials and, consequentially, …), we agree there is no clear explanation, however we thought that this result would be interesting per se.

Reviewer #3 (Recommendations for the authors):– For better clarification of the nature of the collected data, the behavioral states of the animals (passive presentation of visual stimuli, head-fixed, etc) should be mentioned in the Dataset section in the Materials and methods.

We updated the first paragraph (“Dataset”) of the “Methods” section accordingly:

“Our analysis was based on the Visual Coding – Neuropixels dataset of head-fixed recordings in awake mice provided by the Allen Institute…” and “Each recording session had a length of ~3 hour. In each experiment a series of visual stimuli were presented in front of the animal (gabors, drifting gratings, static gratings, natural images, movies, flashes). Mice did not undergo any training associated with these stimuli.”

– As described in the public review, the authors can provide more thoughts on the physiological impacts and or a particular role of the directional propagation of the ripples. However, this suggestion is optional.

We added a new paragraph in relation to figure 5 at the end of the discussion:

“Functional differences between ripples are supported by our finding that the ripple…“

– Kumar and Deshmukh report stronger ripple propagation along the proximodistal axis. I am wondering how much of the present finding is explained by the difference between the proximal vs distal CA1 because I see the Neurapixels data from the intermediate CA1 is dominated by the proximal part (Figure 1E). Additional analysis of differentially characterizing the ripple propagation along the proximodistal axis would strengthen the conclusion.

To address this question, we employed “partial correlation” in order to understand the degree of association between two variables, while controlling for the influence of other variables. Partial correlation measures the strength and direction of the linear relationship between two variables after removing the effects of one or more additional variables that may be affecting the correlation. In our case we measured the partial correlation between the ripple lags and the position on the antero-posterior (A-P) axis while controlling for the position on the medio-lateral (M-L) axis. The A-P axis did not explain a substantial amount of variability in lag (R² medial = 0.0037 , p-value = 4.48e-01, R² lateral = 0.0698, p-value = 1.74e-03), the opposite was true for the M-L axis (while controlling for the A-P axis): R² medial = 0.6648 (p-value = 2.20e-38), R² lateral = 0.4989 (p-value = 3.75e-22).

We added this result in the text:

“and the antero-posterior (A-P) axis did not explain a substantial amount of variability in lag…”

and added the explanation of the method in the method section.

References

Henriksen, E. J., Colgin, L. L., Barnes, C. A., Witter, M. P., Moser, M. B., and Moser, E. I. (2010). Spatial representation along the proximodistal axis of CA1. Neuron, 68(1), 127-137. https://doi.org/10.1016/j.neuron.2010.08.042

Strange, B. A., Witter, M. P., Lein, E. S., and Moser, E. I. (2014). Functional organization of the hippocampal longitudinal axis. Nat Rev Neurosci, 15(10), 655-669. https://doi.org/10.1038/nrn3785